# Computational design of thermostabilizing point mutations for G protein-coupled receptors

Petr Popov[1,2], Yao Peng[3], Ling Shen[3,4], Raymond C Stevens[1,3,5,6], Vadim Cherezov[1,2,5,6], Zhi-Jie Liu[3,4,7], Vsevolod Katritch[1,2,5,6]*

[1]Department of Biological Sciences, University of Southern California, Los Angeles, Los Angeles, United States; [2]Moscow Institute of Physics and Technology, Dolgoprudny, Russia; [3]iHuman Institute, ShanghaiTech University, Shanghai, China; [4]School of Life Science and Technology, ShanghaiTech University, Shanghai, China; [5]Department of Chemistry, University of Southern California, Los Angeles, Los Angeles, United States; [6]Bridge Institute, University of Southern California, Los Angeles, Los Angeles, United States; [7]Insitute of Molecular and Clinical Medicine, Kunming Medical University, Kunming, China

**Abstract** Engineering of GPCR constructs with improved thermostability is a key for successful structural and biochemical studies of this transmembrane protein family, targeted by 40% of all therapeutic drugs. Here we introduce a comprehensive computational approach to effective prediction of stabilizing mutations in GPCRs, named CompoMug, which employs sequence-based analysis, structural information, and a derived machine learning predictor. Tested experimentally on the serotonin 5-HT$_{2C}$ receptor target, CompoMug predictions resulted in 10 new stabilizing mutations, with an apparent thermostability gain ~8.8°C for the best single mutation and ~13°C for a triple mutant. Binding of antagonists confers further stabilization for the triple mutant receptor, with total gains of ~21°C as compared to wild type apo 5-HT$_{2C}$. The predicted mutations enabled crystallization and structure determination for the 5-HT$_{2C}$ receptor complexes in inactive and active-like states. While CompoMug already shows high 25% hit rate and utility in GPCR structural studies, further improvements are expected with accumulation of structural and mutation data.
DOI: https://doi.org/10.7554/eLife.34729.001

*For correspondence:
katritch@usc.edu

## Introduction

G-protein coupled receptors (GPCRs) represent the largest family of transmembrane proteins, which is involved in regulation of all major physiological functions and comprises more than 25% of established therapeutic targets (*Lagerström and Schiöth, 2008*; *Rask-Andersen et al., 2014*). However, high conformational flexibility and low thermostability of these receptors have always presented major challenges for their structural, biophysical, and biochemical characterization. With exception of visual rhodopsin, structural characterization of all other 50 GPCRs so far required substantial efforts in protein engineering to design GPCR constructs suitable for crystallization (*Cherezov et al., 2007*; *Warne et al., 2009*; *Chun et al., 2012*; *Katritch et al., 2013*; *Stevens et al., 2013*; *Pándy-Szekeres et al., 2018*). The design typically involves truncations of N- and C- termini, replacements of flexible loops and/or termini with soluble fusion domains (*Chun et al., 2012*), stabilizing co-crystallization partners (*Zhang et al., 2015*), and in many cases introduction of one or more point mutations (reviewed in (*Heydenreich et al., 2015*)).

**eLife digest** The trillions of cells in the human body rely on receptors that sit in their cell membranes to communicate with each other. Hundreds of different receptors belong to the G protein-coupled receptor superfamily (called GPCRs for short) and play vital roles in the all organs and bodily systems. Indeed, GPCRs are the targets for almost 40% of therapeutic drugs. As such, deciphering the shape and activity of GPCRs is key to understanding the normal workings of the human biology and could help scientists discover new treatments for various diseases, from depression to high blood pressure to cancer. These receptors, however, are notoriously flimsy and unstable, making them difficult to work with in the laboratory.

Different approaches have been developed to make GPCRs more stable, usually by swapping one or a few of the amino acid building blocks in the protein for other amino acids. Currently, this requires a costly and slow trial-and-error approach in which each amino acid out of 300-400 in the protein is mutated and tested experimentally.

To speed up and reduce the cost of the process, Popov et al. asked if a computer could predict which mutations in the protein would stabilize it, meaning that fewer proteins would actually need to be tested. Four computer algorithms based on four different principles were developed and verified. The first one compares the target GPCR to other closely related receptors, trying to detect variations that cause the instability. The second tries to build in specific stabilizing interactions, or "bridges", between different parts of the receptor. The third algorithm searches the known structures of other GPCRs for useful mutations. Finally, the fourth one uses accumulated data on the stability of hundreds of mutations in different GPCRs to train a machine learning predictor to recognize stabilizing mutations.

All four algorithms produced useful predictions in a real-life project. Indeed, when combined in one computational tool, named CompoMug, the algorithms made it possible to detect optimal mutations in a human GPCR called 5-HT$_{2C}$. This made the protein much easier to work with in the laboratory, and ultimately helped to solve its three-dimensional structure (which was reported in a separate study, published earlier in 2018)

The 5-HT$_{2C}$ receptor is involved in regulating, among other things, mood and appetite. Details of its structure might therefore help researchers to design new antidepressants and obesity treatments. Moreover, CompoMug is already helping structural biologists to solve the structures of other GPCRs, which will further facilitate many aspects of GPCR drug discovery.

DOI: https://doi.org/10.7554/eLife.34729.002

Point mutations have shown to be especially important for thermostabilizing GPCR and making them amenable for structure-based drug design applications, which involve receptor co-crystallization with typically low-affinity hit or lead compounds. For example, point mutations were used to thermostabilize β$_1$-adrenergic (ADRB1) and A$_{2A}$ adenosine (A$_{2A}$AR) receptors in both agonist and antagonist bound states, resulting in more than a dozen structures for each receptor including co-crystals with ligands in a micromolar affinity range (*Moukhametzianov et al., 2011*; *Christopher et al., 2013*; *Warne et al., 2008*; *Warne et al., 2011*). In the case of thermostabilized A$_{2A}$AR, structural and biophysical characterization of initial hits led to structure-based discovery and optimization of preclinical candidates for Parkinson disease (*Langmead et al., 2012*). Moreover, thermostabilized GPCR constructs can streamline biochemical characterization of ligand binding in sensor-based high-throughput screening (HTS) (reviewed in (*Kumari et al., 2015*)) and measurements of ligand-binding kinetics by surface plasmon resonance (SPR) (*Christopher et al., 2013*; *Congreve et al., 2011*; *Rich et al., 2009*).

However, currently employed experimental identification of stabilizing mutations by alanine scanning (*Errey et al., 2015*) or directed evolution approaches (*Egloff et al., 2014*; *Schlinkmann et al., 2012*) is a very resource consuming process, and only a few GPCRs have been successfully stabilized so far (reviewed in (*Vaidehi et al., 2016*)). Furthermore, stabilizing mutations obtained by these methods have shown very limited transferability between different GPCRs (*Heydenreich et al., 2015*; *Serrano-Vega and Tate, 2009*), requiring extensive stabilization campaigns to be performed for each individual receptor.

Computational approaches could provide a cost- and time-effective alternative for GPCR stabilization. The already existing in silico prediction tools for soluble proteins (*Kumar et al., 2006*; *Khan and Vihinen, 2010*), however, are not effective for GPCRs because they do not take into account peculiarities of the 7-transmembrane (7TM) nature of the receptors. At the same time, although some of the recently developed GPCR-specific methodologies can be successful in explaining known experimentally-derived mutations (*Vaidehi et al., 2016*; *Bhattacharya et al., 2014*), their success in prediction of new stabilizing mutations has been limited so far, and has not resulted yet in successfully solved crystal structures of new GPCRs.

In this study, we present a set of complementary approaches for predicting stabilizing mutations in GPCRs combined into a CompoMug tool (**COM**putational **P**redictions **O**f **MU**tations in **G**PCRs). CompoMug consists of four modules: knowledge-based, sequence-based, structure-based, and machine-learning-based, taking maximum advantage of accumulated structural and biophysical data. We applied CompoMug to identify stabilizing point mutations for the 5-HT$_{2C}$ receptor, which is an important pharmacological target for the treatment of obesity and neuropsychiatric disorders. Experimental assessment showed that 10 out of the 39 predicted mutations improved stability of the receptor by more than 1.5°C, and one mutation resulted in increase of the apparent melting temperature by up to ~8.8 ± 1.3°C, which is among the highest reported improvements in thermostability by a single point mutation in GPCRs. Moreover, combinations of two or three mutations led to even higher thermostability gains, some of which were compatible with both agonist and antagonist binding. Finally, the mutants predicted by CompoMug allowed for the determination of two 5-HT$_{2C}$ crystal structures in both agonist-bound and antagonist-bound complexes. The CompoMug provides a computational platform for thermostabilization of other GPCRs and can be further evolved with an accumulation of experimental mutation data.

## Computational methods

CompoMug consists of four modules: knowledge-based, sequence-based, structure-based, and machine-learning-based, - combining several approaches to compose a list of the candidate point mutations, which can improve the stability of a GPCR. The general workflow of CompoMug is presented in *Figure 1*. Below we describe each of the modules in details.

### Knowledge-based module

The knowledge-based module employs a short list of established point mutations that have been already shown to improve stability and helped to solve structures for multiple GPCRs. Although in general stabilizing point mutations are not transferable across different GPCRs (*Serrano-Vega and Tate, 2009*), several specific mutations located in structurally or functionally conserved sites have shown increased chances to be beneficial for multiple receptors. Such known point mutations, listed in *Table 1*, could be good candidates for new GPCR targets, even for those with relatively low homology to solved GPCRs. For example, the mutation of a residue in position 3.41 to Trp (X$^{3.41}$W, where X stands for any residue, and superscript shows GPCRdb numbering (*Isberg et al., 2015*)), first identified in the β$_2$ adrenergic receptor (*Roth et al., 2008*), has been now tested in more than 20 receptors by the GPCR Network (*Stevens et al., 2013*) and has shown to increase stability for several of them, helping to solve crystal structures for at least six receptors so far. The list also includes mutations that target residues in the sodium binding pocket, e.g. D$^{2.50}$N, S$^{3.39}$A, and D$^{7.49}$N mutations (*Kruse et al., 2012*; *Fenalti et al., 2014*; *Katritch et al., 2014*). Sodium ions play an important role in class A GPCR signaling (*Katritch et al., 2014*), and, therefore, modifications in the sodium-binding site, e.g. by D$^{7.49}$N mutation, can decouple ligand binding from conformational changes in the intracellular side of the receptor (*Katritch et al., 2014*; *Massink et al., 2015*). Such decoupling apparently reduces conformational heterogeneity of the receptors, resulting in thermostabilization of some receptors, like A$_{2A}$AR (*White et al., 2018*), and facilitating their structure determination, especially in complexes with agonists (see *Table 1*). Note, that while currently only a few mutations in class A can be classified as transferrable 'knowledge-based', the list may continue to grow with an accumulation of additional knowledge on mutations, and also expand to include specific transferrable mutations in other GPCR classes. Algorithmically, we implemented the knowledge-based module as a simple procedure, which checks mutations from *Table 1*, and assigns score

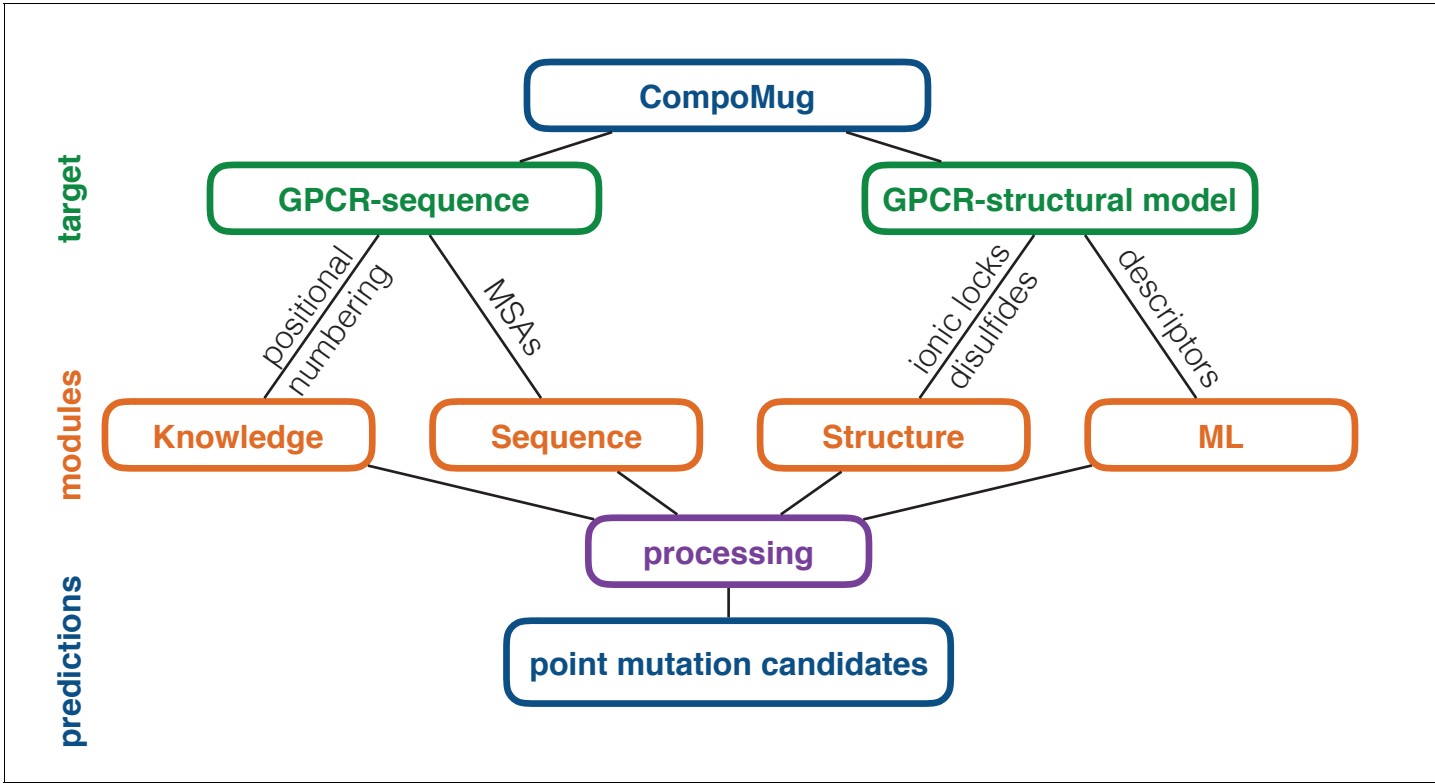

**Figure 1.** CompoMug architecture. The method comprises four modules to predict stabilizing point mutations. The knowledge-based and sequence-based modules operate with only the sequence information about the target receptor, while the structure-based and machine-learning-based modules operate with the structural information. ML – machine learning; MSAs – multiple sequence alignments.

DOI: https://doi.org/10.7554/eLife.34729.003

1.0 if the mutation is potentially applicable (i.e. the wild type residue in the target GPCR corresponds to a residue in *Table 1*), and 0.0 otherwise.

## Sequence-based module

The sequence-based module looks for residues of the target receptor that deviate from a standard conservation pattern in an evolutionarily related group of GPCRs, e.g. receptor orthologs, a subfamily or a branch of the GPCR tree. We hypothesized that such residues in GPCRs are more likely to be destabilizing, and restoring conserved amino acids in such positions might result in receptor

**Table 1.** Knowledge-based transferable point mutations in Class A.

| Position | Mutation | Role | Receptor (PDB ID) |
|---|---|---|---|
| 2.50 | D->N | Sodium pocket | AA2AR (5WF5) |
| 3.39 | S->A | Sodium pocket | AA2AR (5WF6) |
| 3.40 | I->V, A | P-I-F microswitch motif | ADRB1 (4BVN), APJ (5VBL) |
| 3.41 | X->W | stabilization of TM3, TM4, TM5 interface | 5HT2B (4IB4), 5HT1B (4IAR), ADRB1 (5A8E), ADRB2 (3NY8), CXCR4 (3ODU), DRD3 (3PBL) |
| 3.49 | D,G->A | DRY motif | FFAR1 (5TZR), NTR1 (4XES) |
| 5.58 | Y->A | Conserved activation microswitch | FFAR1 (5TZR), ADRB1 (4BVN) |
| 6.37 | L->A | Interferes with DRY motif function | AA2AR (5IU4), NTR1 (4GRV) |
| 7.49 | D->N | Sodium pocket | P2RY1 (4XNV), P2Y12 (4PXZ) |

X = any residue

DOI: https://doi.org/10.7554/eLife.34729.004

stabilization. In CompoMug, the 'deviation score' for an amino acid residue is estimated based on multiple sequence alignment (MSA) of evolutionary related homologous sequences:

$$Score_k^{aa} = \frac{C_k^{max} - C_k^{aa}}{N_{MSA}} - \frac{C_k^{aa}}{C_k^{max}}, \qquad (1)$$

where $N_{MSA}$ is the total number of sequences in the MSA, $C_k^{max}$ is the number of sequences with the most conserved amino acid residue at the position $k$, and $C_k^{aa}$ is the number of sequences that have the same residue $aa$ as the target sequence in this position. As one can see from *Equation 1*, the first term is the highest when the target sequence has the most infrequent amino acid in the position $k$, that is, it approaches 1, when $C_k^{aa} = 1$ and $C_k^{max} \approx N_{MSA}$. The second term penalizes the position $k$ if it lacks a dominating conserved amino acid at the position, that is, the penalty is increased as $C_k^{max}$ is decreased. The total score varies from -1.0 to 1.0, where maximum score 1.0 is ascribed to a deviating amino acid in a super-conserved position (e.g. x.50 in GPCRs). In other words, the preference is given to mutations of rarely observed amino acids in the otherwise highly conserved positions. *Figure 2A and B* schematically show the score computation given an MSA. Apparently, any conservation-related score depends on the set of sequences used to construct the MSA. For example, orthologs share very high sequence similarity with respect to the target GPCR resulting in a few, but usually very clear deviation patterns. On the other hand, comparison with GPCR sequences from different branches has a much more complex conservation pattern that may result in many false positive candidates. To capture the sequence deviations at different levels of GPCR hierarchy, we composed several sets of sequences to construct various MSAs. Specifically, we used five MSAs: (1) ortholog sequences corresponding to the species variations of the target receptor, (2) sequences corresponding to the common sub-family (sequence identity for the TM regions >40%), (3) sequences corresponding to the common GPCR branch (sequence identity for the TM regions >30%), (4) sequences corresponding to the whole non-olfactory class A GPCR (*Rios et al., 2015*), and (5) sequences corresponding to the crystallized receptors. MSAs were generated with the structure-based alignment tool of the GPCRdb (*Isberg et al., 2015*), and in case of the whole class A alignment updated using MAFFT software (*Katoh and Frith, 2012*).

Although the last MSA is not directly related to the evolutionary variation, it may contain information relevant for the GPCR stability and propensity for crystallization. At the same time, the MSA for whole class A GPCR would capture rare variations in the most conserved residue positions of class A, including $N^{1.50}$, $D(E)R^{3.50}Y$, $FxxxCWxP^{6.50}$ and $NP^{7.50}xxY$. Given all five MSAs, we computed positional scores for each MSA, as well as the global score as the average of the individual MSA scores. In a special case of non-conserved Gly residues, we multiplied the 'deviation score' by factor of 2, to account for Gly usually destabilizing effect on α-helical secondary structure in the transmembrane helices of the receptor. *Figure 2* schematically shows the workflow of the sequence-based module applied to the 5-HT$_{2C}$ receptor.

## Structure-based module

The structure-based module is focused on identifying pairs of residues, which could form a salt bridge (also called ionic lock) when replaced with charged amino acids, or disulfide bonds when replaced with cysteines. Such ionic locks and covalent bonds can help to restrict the conformational flexibility of the receptor and improve stability. A successful use of the structure-based approach requires an accurate 3D structural model, which can be derived based on the close homology with a known crystallographic structure. In this study, structural models were obtained using the template-based homology modeling implemented in ICM-Pro v.3.8 molecular modeling suite (molsoft.com), followed by the backbone regularization and exhaustive Monte-Carlo side-chain refinement in internal coordinates. To predict potential ionic locks in the structural model, the search is performed for pairs of residues that satisfy the following criteria: i) residues are separated in sequence by at least five residues to exclude pairs of residues belonging to the same α-helix, ii) side chains point toward each other and do not point to the lipid membrane, iii) residue's C$_\beta$-C$_\beta$ distance lies in the range from 7.0 Å to 10.0 Å, and iv) mutations of residues to at least one of four charged pairs (E-K, E-R, D-K, D-R) improve predicted free energy of the receptor after thorough local conformational optimization of the mutants (*Equation 2*)

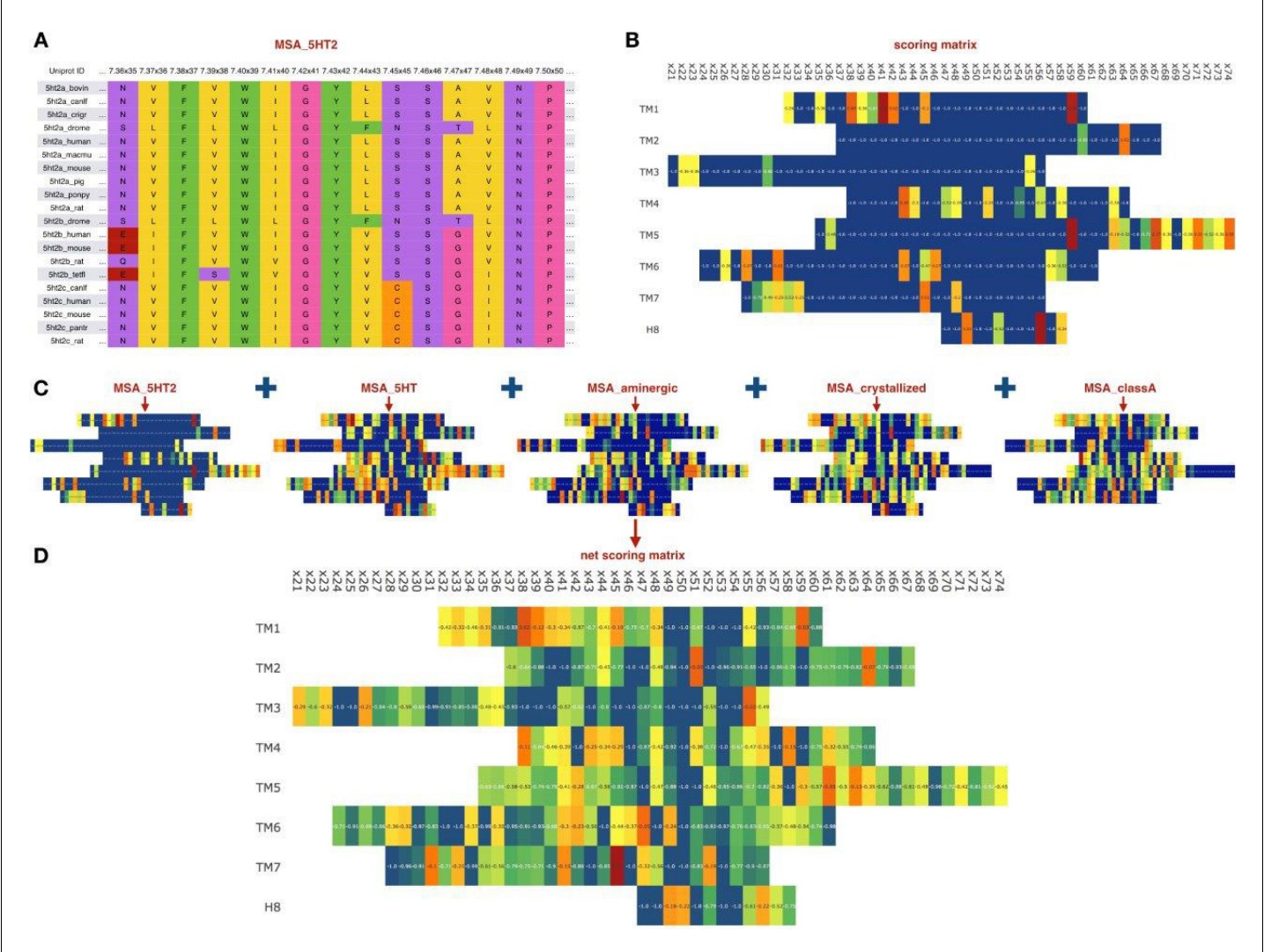

**Figure 2.** Sequence-based module. (**A**) Example of an MSA for orthologs of 5HT2 receptors, residues colored according to their chemical properties. (**B**) Computed scoring matrix from the MSA_5HT2 in the sequence-based module (**C**) Example of the scoring matrices for five different MSAs. (**D**) Combined net scoring matrix. Each position is colored according to the score, from blue (minimal score) to red (maximal score).

DOI: https://doi.org/10.7554/eLife.34729.005

$$E_{folded}^{mut} - E_{unfolded}^{mut} < E_{folded}^{wt} - E_{unfolded}^{wt} \qquad (2)$$

We used energy calculation implemented in the Molsoft ICM-Pro v.3.8. software (molsoft.com). The structural model of the mutant type was obtained by mutation of a given residue followed by Monte Carlo sampling of the flexible side chains for the mutated residue and the neighboring residues. Then the free energy of the unfolded and folded states for the wild and mutant types was approximated by a sum of the empirically derived residue-specific energies.

In order to predict stabilizing disulfide bonds in the receptor, we first employed the DbD software (***Craig and Dombkowski, 2013***) to obtain the initial list of candidates. DbD scans all pairs of residues in a protein and selects those that satisfy geometrical parameters of the disulfide bond, when replaced with cysteines. The geometrical parameters, e.g. $\chi 3$ angle and $C_\beta - C_\beta$ distance, were obtained from analysis of protein structures in PDB. Given the DbD predictions, the final list of candidates was derived using the energy criterion implemented in ICM-Pro (see ***Equation 2***). ***Figure 3*** schematically represents the structure-based module.

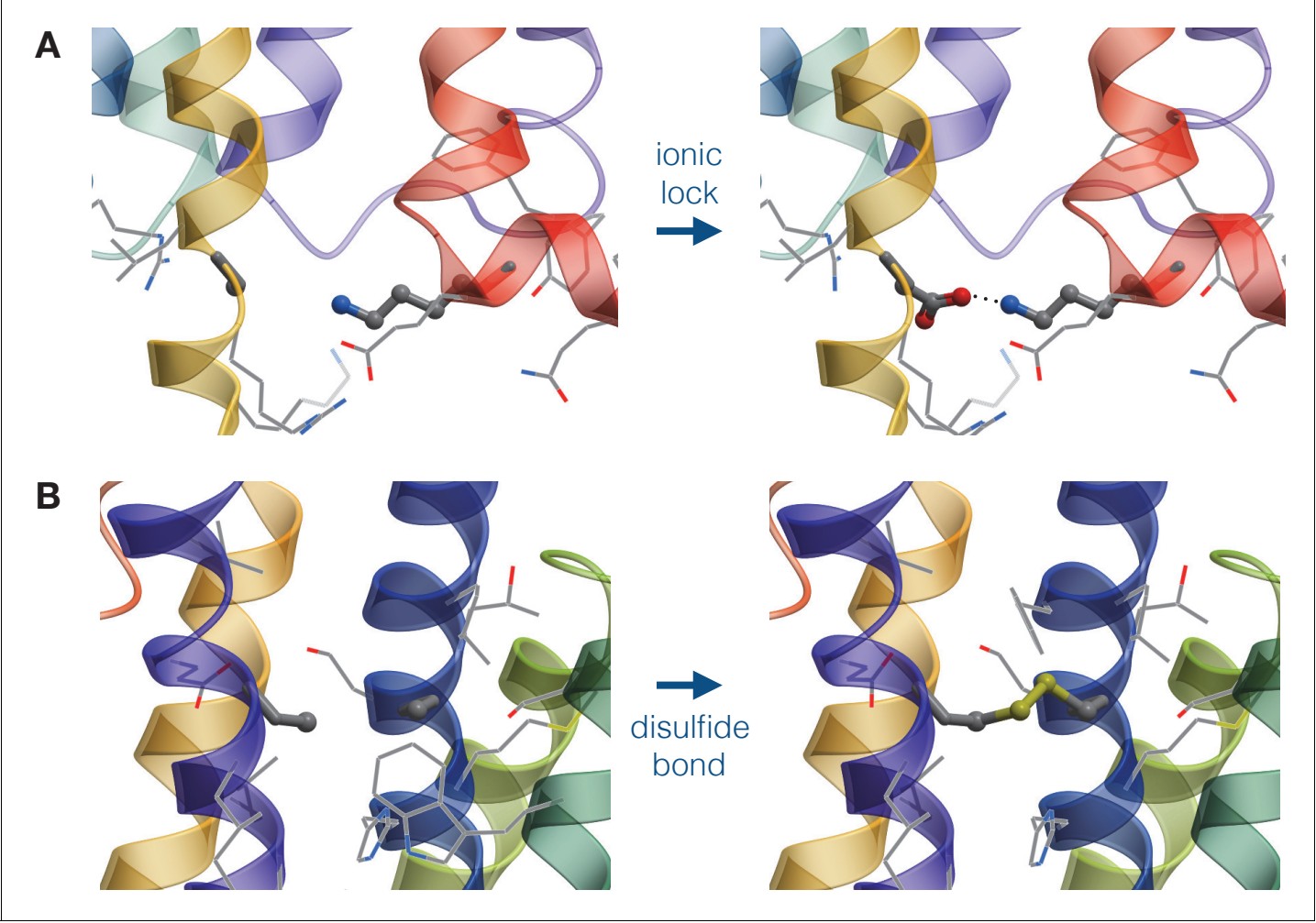

**Figure 3.** Schematic representation of mutations generated by the structure-based module. (**A**) Design of an Asp-Lys ionic lock by the point mutation of an Ala residue. (**B**) Design of a disulfide bridge by the double mutation of Ala residues..

DOI: https://doi.org/10.7554/eLife.34729.006

## Machine learning module

With the accumulation of experimental data on the stability of GPCR mutants, it becomes feasible to derive powerful prediction models using machine learning techniques. Our prediction model is derived using (i) a *training benchmark,* composed from site-specific mutations performed on GPCRs with known structure, (ii) a *feature vector*, consisting of structure-based and energy-based descriptors, which reflect important changes in the protein upon a point mutation, and (iii) a *support vector machine* method as implemented in the *libsvm* package (*Chang and Lin, 2011*). Each of these steps is described below in details.

### Training benchmark

To compose the training benchmark we used available alanine scanning mutagenesis data for three GPCR receptors: neurotensin receptor NTS1 (*Shibata et al., 2009*), $A_{2A}$ adenosine receptor (*Magnani et al., 2008*), and $\beta_1$ adrenergic receptor ADRB1 (*Serrano-Vega et al., 2008*). Point mutations that improve thermostability of these receptors were used as positive examples, while reverse mutations were used as negative examples for training. Further, in order to expand the training benchmark, we considered the remaining alanine mutations, that is, those which were not reported as stabilizing, as negative examples. It is worth to note that such assumptions may introduce some false negative examples into the training set, because some of the alanine mutations were filtered

out due to the lower expression level, rather than due to a decrease in the receptor stability. Overall, the training benchmark consists of 79 stabilizing point mutations and 923 non-stabilizing point mutations.

## Feature vector

Given the training set, we projected each point mutation as a vector onto a feature space, where the coordinates of the feature vector encode information relevant to a change in the receptor stability upon introducing the point mutation. To compose a feature vector, we used characteristics of three different types. Namely, for wild type and mutated residues we used sequence-based characteristics, which could be extracted from the primary structure of the protein (hydrophobicity, polarity, charge, side chain volume, solvent-accessible area, polarizability), structure-based characteristics, which could be extracted from the secondary and the tertiary structures of the protein (number of polar, charged, hydrophobic, and aromatic contacts, residue exposure, contact area, void volume, relative accessible solvent area), and energy-based characteristics, which could be extracted from the tertiary structures of the protein given the force-field (potential of mean force, electrostatic, van der Waals, solvation, hydrogen bond, and total energies). To obtain a structural model of a mutant type we mutated a given residue and performed Monte Carlo minimization with flexible side chains of the mutated residue and its neighboring residues, keeping the rest of the receptor rigid, using the Molsoft ICM-Pro v.3.8. software (molsoft.com). To calculate components of the feature vector we used built-in functions of ICM-Pro.

## Support Vector Machine classifier

Feature vectors computed for each point mutation in the training benchmark are then combined into the feature matrix. Each entry in the feature matrix is labeled with +1, if the corresponding point mutation stabilizes receptor, or −1 otherwise. Given this mapping of point mutations into the feature space, one can construct a hypersurface which separates +1 feature vectors from −1 feature vectors, using the support vector machine (SVM) approach. We used the libsvm package libraries (*Chang and Lin, 2011*) to accomplish this task. There are two free parameters in this classification problem. Namely, the regularization parameter C, which is a tradeoff between the misclassification and 'smoothness' of the separating hypersurface, and the kernel parameter γ, which corresponds to the variance of the radial basis function. Optimal values for these parameters were defined using the two-fold cross-validation procedure (see below). *Figure 4* schematically represents the machine-learning-based module.

## Cross-validation of the machine learning classifier

To validate the machine learning classifier the data set was randomly split into two parts: the training part, which consists of 65% of the training set, and the validation part, which consists of the remaining 35%. Each subset was adjusted to retain the ratio of the +1 feature vectors, corresponding to stabilizing point mutations, and −1 feature vectors, corresponding to non-stabilizing point mutations. To optimize the performance of the algorithm, we scanned values of the two free parameters (C and γ) on a grid [0.0, 50.0]×[0.0, 50.0] with a step size of 0.2 for both parameters, thus, yielded 250*250 = 62,500 different prediction models. Then for each prediction model, we calculated the positive predictive value (PPV), which is the ratio of true positive rate (TPR) and the sum of true positive and false positive rates (FPR) predictions:

$$PPV = \frac{TPR}{TPR + FPR} \tag{3}$$

We repeated this procedure 10 times and scored parameters, based on the rank of each pair (rank one corresponds to the maximum *PPV*) in each cross-validation run:

$$Score(C, \gamma) = \sum_{j=1}^{10} \frac{Rank(PPV_i^j)}{max_i Score(C, \gamma)_i} \tag{4}$$

We identified four pairs of parameters C and γ in different grid regions, which resulted in

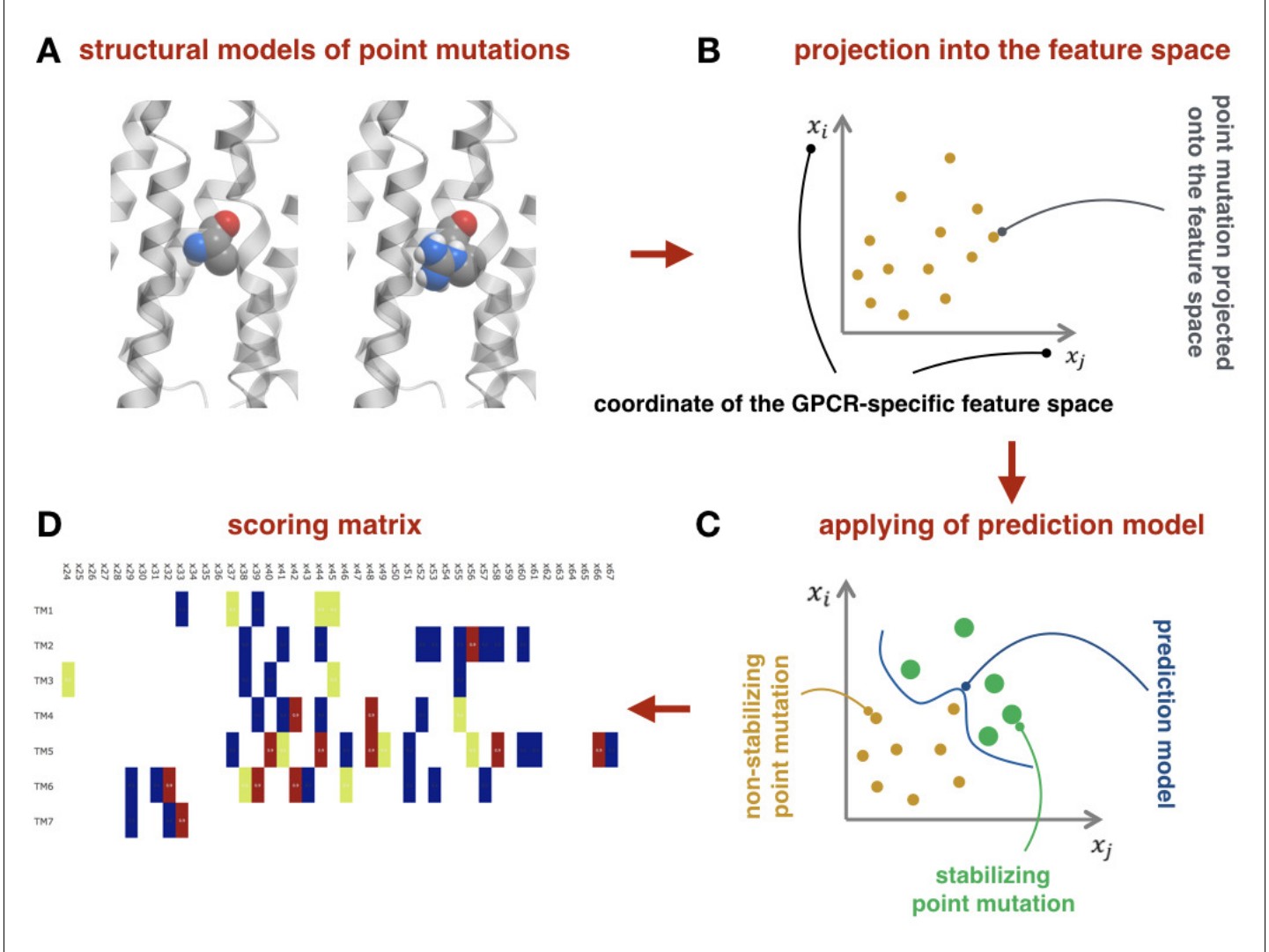

**Figure 4.** Machine-learning-based module. **(A)** Example of structural models for the wild-type and mutant-type receptor. **(B)** Schematic representation of the point mutations mapped into the feature space. **(C)** Schematic representation of the prediction model as the separation curve in the feature space. **(D)** The net scoring matrix calculated with respect to the weights of the prediction models (blue and red colors correspond to the lowest and highest scores, respectively).

DOI: https://doi.org/10.7554/eLife.34729.007

approximately the same low score, but provided different expected true positive and false negative rates (see *Figure 5*).

Finally, we re-derived four prediction models on the whole training set using the obtained parameters C and γ. The estimated TPR and FPR for each derived prediction model are presented in *Figure 5B*. As one can see we chose prediction models with different TPR and FPR. The reason for that is to control the number of output predictions (note that 'all −1' model, that is, model that treats all point mutations as non-stabilizing, has perfect FPR, while 'all +1' model, that is, model that treats all point mutations as stabilizing has perfect TPR). For example, the first prediction model outputs only a few predictions, but with high confidence, in contrast to the fourth prediction model, which outputs more predictions, but also increases the number of false positive ones. The total score of a point mutation is, thus, weighted according to the prediction model.

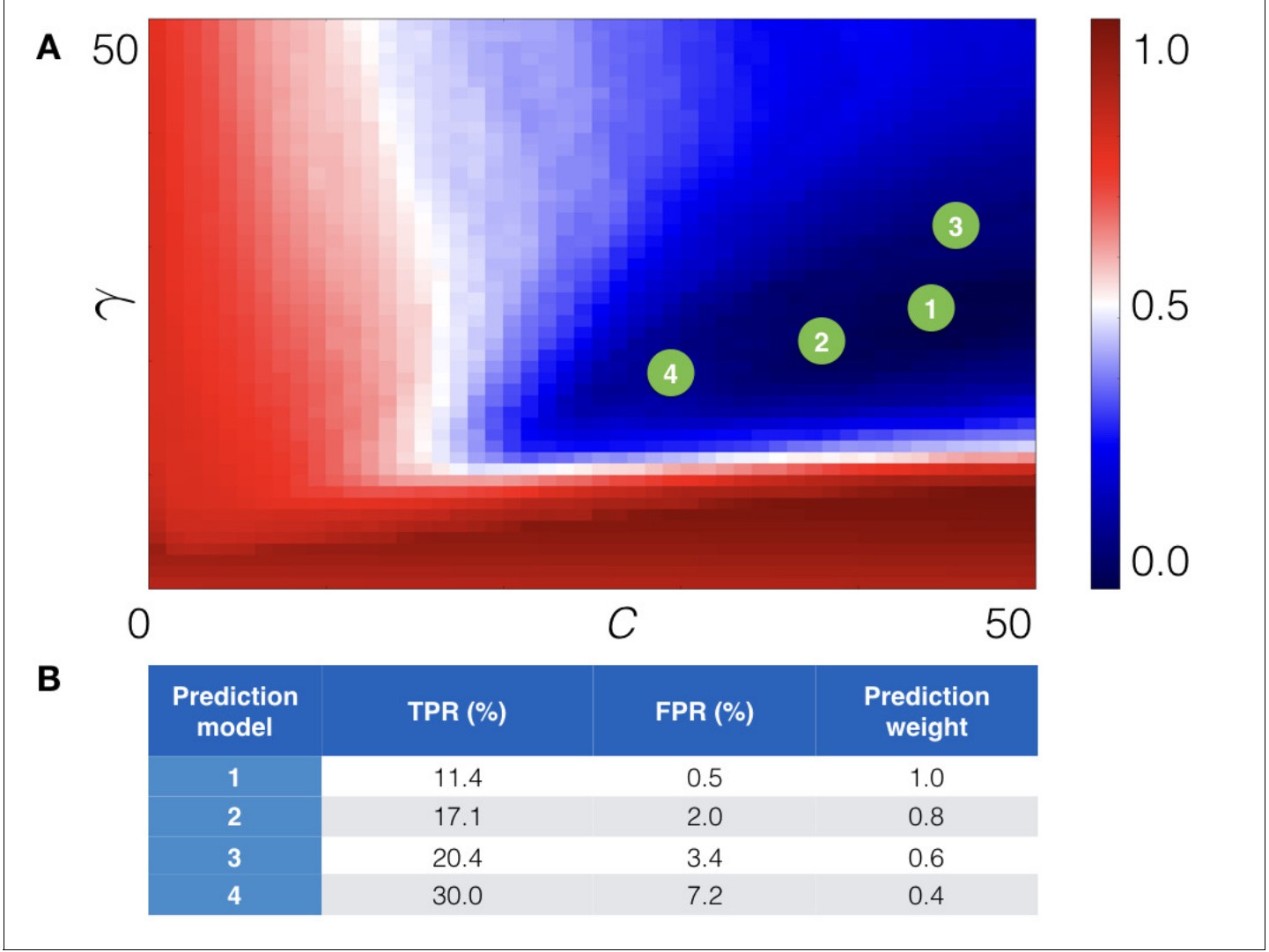

**Figure 5.** Cross-validation of the machine learning module. (**A**) The cross-validation grid for the parameters C and γ. Pairs of C and γ of the top 4 prediction models are depicted with green numbered circles. (**B**) Estimated true (TPR) and false (FPR) positive rates for the derived prediction models along with prediction weight, which is added to score of a point mutation.

DOI: https://doi.org/10.7554/eLife.34729.008

## Post processing

Given the output predictions from each module, we then filtered out point mutations that may affect ligand binding. For this purpose, we analyzed GPCR-ligand interactions in solved GPCR structures (*Munk et al., 2016*) and excluded residue positions that appear in the binding pocket in more than five different class A GPCR structures. We also did not consider predictions in the less conserved regions that lack secondary structure, e.g. loops and N/C – termini, since the modeling accuracy for these regions is much lower, compared to the transmembrane alpha-helical core.

## Experimental methods

### Protein construct

The sequence of the human 5-HT$_{2C}$R gene was synthesized by GenScript. The modified thermostabilized apocytochrome b$_{562}$RIL (BRIL) as a fusion partner was inserted into the receptor's third intracellular loop (IL3) at L246 and M300 of the human 5-HT$_{2C}$R gene, using overlapping PCR. The construct was further optimized by truncation of N-terminal residues 1–39 and C-terminal residues 393–458.

The $\Delta$N-5-HT$_{2C}$-BRIL-$\Delta$C DNA was subcloned into a modified pFastBac1 vector for expression in *Spodoptera frugiperda* (*Sf9*) cells. The chimera sequence has a haemagglutinin (HA) signal sequence followed by a FLAG tag at the N-terminus, a PreScission protease site, and a 10 × His tag at the C-terminus. Rationally designed point mutations were engineered into the 5-HT$_{2C}$R gene by standard QuickChange PCR.

## Protein expression in Bac-to-Bac baculovirus expression system

The Bac-to-Bac Baculovirus Expression System (Invitrogen) was used to generate high-titer recombinant baculovirus (>$10^9$ viral particles per ml). Recombinant baculovirus was produced by transfecting recombinant bacmids (2.5–5 µg) into *Spodoptera frugiperda* (*Sf9*) cells (2.5 ml, density of $10^6$ cells per ml) using 5 µl of X-tremeGENE HP DNA Transfection Reagent (Roche) and Transfection Medium (Expression Systems). After 4 d of shaking at 27°C, P0 viral stock (~$10^9$ virus particles per ml) was harvested as the supernatant of the cell suspension to produce high-titer viral stock. Viral titers were analyzed by flow cytometry on cells stained with gp64-PE antibody (Expression Systems). 5-HT$_{2C}$R was expressed by infecting *Sf9* cells at a cell density of 2–3 × $10^6$ cells per ml with P1 virus at MOI (multiplicity of infection) of 5. Cells were harvested by centrifugation 48 hr post infection and stored at −80°C for future use.

## Protein purification

Thawed insect cells were disrupted in a hypotonic buffer containing 10 mM MgCl$_2$, 20 mM KCl, 10 mM HEPES (pH 7.5) and EDTA-free complete protease inhibitor cocktail tablets (Roche). The isolated raw membranes were extensively washed by twice repeated centrifugation at 40000 rpm for 30 min at 4°C in the same hypotonic buffer. Subsequently, soluble and membrane associated proteins were removed in a high osmotic buffer containing 10 mM MgCl$_2$, 20 mM KCl, 1.0 M NaCl, 10 mM HEPES (pH 7.5) and EDTA-free complete protease inhibitor cocktail tablets (three times). Purified membranes were flash-frozen in liquid nitrogen and stored at −80°C for further use. Purified membranes were thawed at room temperature and incubated in the presence of 50 µM ligand and protease inhibitor cocktail at 4°C for 2 hr. The membranes were incubated with 1.0 mg/ml iodoacetamide (Sigma) for 30 min and were solubilized in the buffer containing 50 mM HEPES (pH 7.5), 1% (w/v) n-dodecyl-beta-D-maltopyranoside (DDM, Anatrace), 0.2% (w/v) cholesterol hemisuccinate (CHS, Sigma-Aldrich) and 150 mM NaCl, at 4°C for 2.5 hr. The solubilized 5-HT$_{2C}$R proteins in the supernatant were isolated by high-speed centrifugation (Beckman), and then incubated at 4°C overnight with TALON IMAC resin (Clontech), 800 mM NaCl and 20 mM imidazole as the final buffer concentration. The resin was washed with 10 column volumes of washing buffer I containing 50 mM HEPES (pH 7.5), 0.1% (w/v) DDM, 0.02% (w/v) CHS, 800 mM NaCl, 10% (v/v) glycerol, 20 mM imidazole, 50 µM ligand (only for the ligand binding case), and six column volumes of washing buffer II containing 50 mM HEPES (pH 7.5), 0.02% (w/v) DDM, 0.004% (w/v) CHS, 500 mM NaCl, 10% (v/v) glycerol and 50 µM ligand (only for the ligand binding case) without imidazole. The protein was eluted using four column volumes of elution buffer containing 50 mM HEPES (pH 7.5), 0.02% (w/v) DDM, 0.004% (w/v) CHS, 500 mM NaCl, 10% (v/v) glycerol, 250 mM imidazole and 50 µM ligand (only for the ligand binding case). The 5-HT$_{2C}$R protein sample was concentrated to ~10 mg/ml using a 100 kDa cutoff concentrator (Sartorius). The protein yield and monodispersity were measured by analytical size-exclusion chromatography, aSEC (Agilent).

## Protein Stability conducted by CPM Assays

Protein thermostability was measured by a microscale fluorescent thermal stability assay as previously detailed (*Alexandrov et al., 2008*). For thermostability assay, CPM (*N*-([4-(7-diethylamino-4-methyl-3-coumarinyl) phenyl] maleimide) dye was dissolved in DMSO at 4 mg/ml as stock solution and diluted 1:20 in buffer (25 mM HEPES, pH 7.5, 500 mM NaCl, 5% (v/v) glycerol, 0.01% (w/v) DDM, 0.002% (w/v) CHS) before use. 1 µl of diluted CPM was added to the same buffer with approximately 0.5–2 µg 5-HT$_{2C}$ receptor in a final volume of 50 µl. The thermal denaturation assay was performed in a Rotor-Gene realtime PCR cycler (Qiagen). The excitation wavelength was 365 nm and the emission wavelength was 460 nm. All assays were performed over a temperature range from 25°C to 95°C using a temperature range rate 2.0 °C/min. The stability data were processed with GraphPad Prism.

## Results

### Limited benchmarking with alanine scanning data

The initial training set benchmarking of the CompoMug prediction algorithms was performed with the alanine scanning data available for neurotensin receptor NTS1 (*Shibata et al., 2009*), adenosine receptor AA2AR (*Magnani et al., 2008*), and $\beta_1$ adrenergic receptor ADRB1 (*Serrano-Vega et al., 2008*; *Heydenreich et al., 2015*). Due to the nature of the experimental data, such comparison is limited to only X to A (where X is any residue) and A to L point mutations, and the benchmark employed only sequence-based and machine learning modules. For each receptor, we kept top 40 predicted single point mutations and compared the results with the experimental alanine data for the three receptors (see *Supplementary file 1*). For the human AA2AR, turkey ADRB1, and rat NTS1 receptors CompoMug successfully predicts 20, 11, and 9 stabilizing mutations out of 39, 18, and 20 reported mutations in the transmembrane region, suggesting about 50% recall rate in this initial benchmark.

### Application of CompoMug to the 5-HT$_{2C}$ receptor

To test the algorithms in a real case of a blind predictions for a new target prospective screening, we applied CompoMug to predict stabilizing point mutations for the serotonin 5-HT$_{2C}$ receptor. The 5-HT$_{2C}$ receptor is widely expressed within the central and the peripheral nervous systems and appears to play a prominent role in psychiatric disorders. Thus, obtaining the structure of this receptor could help for better understanding and treatment of the pathophysiology of obesity and psychiatric disorders including schizophrenia, anxiety, and depression (*Wacker et al., 2013*)(*Peng et al., 2018*).

To select candidates for point mutations we used the knowledge-based, sequence-based, structure-based and machine-learning modules of CompoMug as described in Computational Methods. In the *sequence-based module,* we composed five different MSAs (see *Supplementary file 2*): orthologs of 5-HT$_{2C}$ receptor, orthologs of all 5-Hydroxytryptamine GPCRs, aminergic receptors (human only), crystallized receptors (class A only), and class A alignment (non-olfactory) (*Rios et al., 2015*). For the *structure-based module,* we first constructed the 5-HT$_{2C}$ homology model based on the structure of the 5-HT$_{2B}$ receptor (PDB ID 4IB4) (*Wacker et al., 2013*). These two serotonin receptor subtypes share 62% of identical residues in the 7TM region (49% for the full sequence). This structural model was also used to generate *239*19 = 4541* models (considering 239 residues in the TM regions and 19 possible amino acid substitutions) with conformationally optimized point mutations as the input for the *machine-learning-based module*, followed by the score assignment with the derived prediction models. After the post-processing procedure, a list of 39 mutations from different modules was selected for experimental testing, as presented in *Table 2* Note, that several mutations were predicted by more than one module.

### Experimental testing of individual CompoMug mutations

A total of 39 mutations predicted by CompoMug (see *Table 2*) were tested on the apo 5-HT$_{2C}$ receptor, using the base construct with N- and C- termini truncations and BRIL fusion as described in Experimental Assays section. The optimal insertion position for BRIL, as well as C- and N-terminal truncations were determined experimentally starting from the WT construct (without mutations), as described in the structural paper (*Peng et al., 2018*). For each point mutation, the receptor was expressed in a modified pFastBac1 vector in *sf9* insect cells, and the aSEC and CPMs profiles were measured for the unliganded receptor (apo) to quantify its thermostability. Point mutations that decreased the receptor expression yield or stability, or for which we could not accurately measure the apparent melting temperature, or did not affect the stability of the protein were disregarded from further experiments. The Tm measurements were repeated for the 10 stabilizing mutations that improved expression and increased apparent melting temperature by at least 1.5°C (**bold** rows in *Table 2*). The most remarkable effect was observed for the C360$^{7.45}$N point mutation predicted with the sequence-based module, which increased the thermostability of the receptor by 8.8 ± 1.3°C in the initial CPM assays. Other mutations showed a moderate effect on thermostability, increasing the apparent melting temperature by 1.5–3.9°C. Six out of ten mutations are substitutions to the hydrophobic residues (A, L, or V), three point mutations are substitutions to the polar or charged residues

**Table 2.** Predicted CompoMug point mutations for 5-HT$_{2C}$ and results of experimental testing.

Mutations shown as **bold** improved aSEC and/or thermostability by more than 1.5 °C ; shown as *italic* had low protein yield or strong aggregation, Tm not measured.

| Mutation | CompoMug module | aSEC* quality | Tm (°C) ± SEM | ΔTm (C) |
|---|---|---|---|---|
| WT | | | 50.4 ± 0.8 | 0.0 |
| I62$^{1.41}$V | Sequence-based | ~ | | −0.7 |
| G69$^{1.48}$A | Sequence-based | - | | −1.4 |
| D99$^{2.50}$N | Knowledge-based | - | | - |
| *H85$^{12.51}$E* | *Structure-based* | *N/A* | | - |
| G103$^{2.54}$A | Sequence-based | - | | −4.4 |
| Y125$^{3.23}$K | Sequence-based | - | | −2.0 |
| Y125$^{3.23}$V | Sequence-based | ~ | | −0.7 |
| M143$^{3.41}$W | Knowledge-based | - | | 0.6 |
| R157$^{3.55}$T | Machine-learning and Sequence-based | - | | −1.8 |
| R157$^{3.55}$Q | Sequence-based | - | | −2.0 |
| T169$^{4.40}$K | Sequence-based | + | | 0.2 |
| **A171$^{4.42}$L** | **Machine-learning** | ~ | 52.3 ± 1.2 | 1.9 |
| I172$^{4.43}$A | Sequence-based | - | | 1.1 |
| I172$^{4.43}$F | Sequence-based | ~ | | 0.6 |
| **G184$^{4.55}$A** | **Machine-learning** | + | 51.9 ± 0.1 | 1.5 |
| N203$^{ECL2}$D | Structure-based | - | | −2.6 |
| F220$^{5.45}$I | Machine-learning | ~ | | 0.0 |
| F224$^{5.48}$Y | Machine-learning and Sequence-based | - | | −3.3 |
| C235$^{5.59}$F | Sequence-based | ~ | | 0.1 |
| *L236$^{5.60}$R* | *Machine-learning and Sequence-based* | *N/A* | | - |
| **V240$^{5.64}$A** | **Sequence-based** | + | 52.4 ± 0.5 | 2.0 |
| V240$^{5.64}$S | Sequence-based | + | | 0.3 |
| G314$^{6.38}$A | Machine-learning-based | - | | −4.0 |
| **L333$^{6.57}$V** | **Machine-learning and Sequence-based** | + | 53.7 ± 0.6 | 3.3 |
| K348$^{7.32}$A | Sequence-based | - | | −4.4 |
| **C360$^{7.45}$N** | **Sequence-based** | + | 59.2 ± 0.5 | 8.8 |
| **G362$^{7.47}$L** | **Sequence-based** | + | 52.3 ± 0.7 | 1.9 |
| **G362$^{7.47}$A** | **Sequence-based** | + | 54.3 ± 0.7 | 3.9 |
| L370$^{7.55}$D | Structure-based | - | | −2.3 |
| K373$^{8.48}$E | Structure-based | - | | −0.4 |
| **I374$^{8.49}$D** | **Structure-based** | + | 53.9 ± 0.8 | 3.5 |
| **I374$^{8.49}$T** | **Sequence-based** | + | 54.1 ± 0.9 | 3.7 |
| Y375$^{8.50}$F | Sequence-based | - | | −2.4 |
| N381$^{8.56}$R | Sequence-based | ~ | | 0.6 |
| *T67$^{1.46}$C/G103$^{2.54}$C* | *Structure-based* | - | | - |
| *V74$^{1.53}$C/A96$^{2.47}$C* | *Structure-based* | - | | - |
| *A87$^{2.38}$C/A171$^{4.42}$C* | *Structure-based* | ~ | | - |
| **A98$^{2.49}$C/A140$^{3.38}$C** | **Structure-based** | ~ | 52.8 ± 1.0 | 2.4 |
| *T369$^{7.54}$C/Y375$^{8.50}$C* | *Structure-based* | *N/A* | | - |

*aSEC quality is denoted as improved (+), unchanged (~), and degraded (-) as compared to the base construct apo receptor.

DOI: https://doi.org/10.7554/eLife.34729.009

(T, N, or D), and one double mutation corresponds to an engineered disulfide bridge (see *Table 2*). We also observed that improvements in aSEC and thermostability were well correlated, meaning that point mutations augmented both aSEC quality and apparent melting temperature.

## Design and testing of combined mutations

After testing single mutations, we devised a list of potentially additive double and triple combinations of point mutations, all of them including the C360$^{7.45}$N mutation. Specifically, we first tested the C360$^{7.45}$N mutation in combinations with all other mutations, as well as double mutation C360$^{7.45}$N-G362$^{7.47}$A in combination with other mutants. These double and triple combinations were tested for the apo receptor and the receptor in complex with different 5-HT$_{2C}$ binding ligands, including an agonist ergotamine and five different antagonists.

As *Figure 6* shows (see *Figure 6—source data 1* for raw data), the tested combinations further improve thermostability of apo receptor, with the maximal observed increase in Tm reaching ~13°C for the triple mutation C360$^{7.45}$N, G362$^{7.47}$A, A171$^{4.42}$L. Moreover, binding of antagonist mesulergine improved thermostability of this triple mutant by additional ~8°C, resulting in a total of 21°C increase in Tm, as compared to the apo base receptor construct. Interestingly, this same triple mutation was destabilized by binding of agonist ergotamine as compared to the apo mutant. In general, while the C360$^{7.45}$N point mutation makes most substantial contribution to the stability of the apo and agonist-bound receptor, the addition of most other point mutations (except for V240$^{5.64}$A)

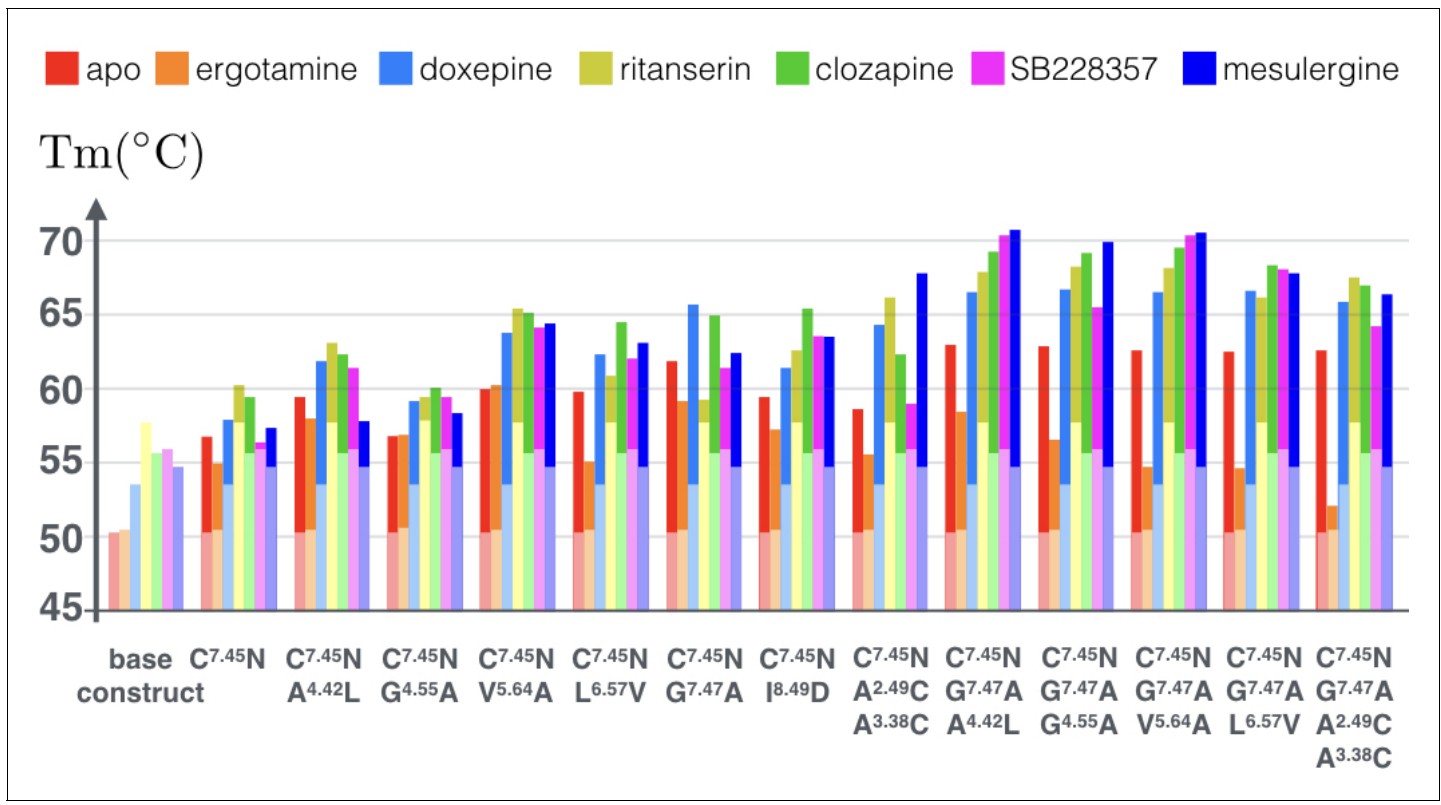

**Figure 6.** Apparent thermostability of 5-HT$_{2C}$ constructs with combined CompoMug mutations. In apo form or in complex with an agonist (ergotamine) or antagonists (doxepin, ritanserin, clozapine, mesulergine, and SB228357). Light colored bars highlight the reference temperatures for the base construct, the full color bars show the additional effect of mutations on these complexes. The expected error for each measurement does not exceed 1.2°C.

DOI: https://doi.org/10.7554/eLife.34729.010

The following source data is available for figure 6:

**Source data 1.** Data for apparent thermostability of 5-HT$_{2C}$ constructs with combined CompoMug mutations and in complexes with ligands, as shown in *Figure 6* (estimated error <1.2°C).

DOI: https://doi.org/10.7554/eLife.34729.011

predominantly stabilizes the antagonist-bound receptor conformation, which was previously less amenable to crystallization. The biggest contrast between agonist and antagonist bound state thermostability (~16°C) was observed for the quadruple mutant construct with an engineered disulfide bond ($C360^{7.45}N$, $G362^{7.47}A$, $A98^{2.49}C/A140^{3.38}C$), suggesting that the introduction of the rigid covalent link between the TM2 and TM3 fixes receptor in the inactive conformational state.

## Predicted mutations enable crystallization and structure determination of 5-HT$_{2C}$ complexes

The predicted stabilizing point mutations made it possible to obtain first crystals of the 5-HT$_{2C}$ receptor in complex with an antagonist, as well as to improve the diffraction of the agonist-bound crystals from >4 Å to <3.0 Å, as described in our recent paper (*Peng et al., 2018*). The predicted mutations were introduced in the context of an available 5-HT$_{2C}$ construct that included optimized fusion partner and N-, C- termini truncations. In this context, multiple combinations of CompoMug-derived mutants resulted in diffracting crystals of the 5-HT$_{2C}$ receptor. At the same time, the single $C360^{7.45}N$ mutation was found as sufficient to solve structures in complex with agonist ergotamine (at 3.0 Å resolution), as well as antagonist ritanserin (at 2.7 Å), which is the first antagonist-bound structure of a serotonin receptor (*Peng et al., 2018*).

## Structural analysis of the predicted thermostabilizing mutations in 5-HT$_{2C}$

Determination of the crystallographic structure of the 5-HT$_{2C}$ receptor (*Peng et al., 2018*) now allows more detailed analysis of the stabilizing nature of the discovered by CompoMug mutations. The mutations were modeled based on the atomic structure of the 5-HT$_{2C}$ receptor as shown in *Figure 7*.

For example, the $A171^{4.42}$ residue, located at the intracellular side of TM4, is surrounded by hydrophobic side chains of $Y90^{2.41}$, $F91^{2.42}$, $I175^{4.46}$, and its replacement with a longer Leu side chain could form more favorable hydrophobic contacts. The $G184^{4.55}$ in the middle of TM4 is exposed to the lipid membrane and does not form any contacts with the side chains, and its replacement with Ala could have a stabilizing effect on the α-helix conformation and more favorable hydrophobic contacts with the lipid environment. The $V240^{5.64}$ residue does not form any specific contacts and it is located close to the membrane intracellular boundary, so the $V240^{5.64}A$ mutation may reduce unfavorable contacts with predominantly charged and polar lipid headgroups in this environment. The $L333^{6.57}$ residue points to the membrane and does not form any specific contacts with the neighboring side chains, and the $L333^{6.57}V$ might improve stability by forming more favorable hydrophobic contacts with lipids.

The $C360^{7.45}$ amino acid is rarely observed at the 7.45 position, and it is known that $N^{7.45}$ plays important role in the sodium coordination as a part of the sodium binding pocket (*Katritch et al., 2014*; *Liu et al., 2012*). Thus, the $C360^{7.45}N$ point mutation restores the conserved residue in the sodium binding pocket and improves the stability of the receptor. Given that this point mutation was necessary to obtain the crystallographic structures of the 5-HT$_{2C}$ receptor in both agonist-bound and antagonist-bound conformations, while $D99^{2.50}N$ was detrimental, the integrity of the sodium binding pocket in 5-HT$_{2C}$ receptor apparently plays an important role for the overall receptor stability. The $G362^{7.47}$ residue is partially exposed to the lipid environment, thus both the $G362^{7.47}L/A$ point mutations improve the stability of the receptor by stabilizing the secondary structure of TM7 and ameliorating hydrophobic interactions with the membrane environment. The $I374^{8.49}$ residue is surrounded by positively charged $K83^{ICL1}$, $K373^{8.48}$, $R376^{8.51}$, and $R377^{8.52}$ side chains, so the $I374^{8.49}D/T$ point mutations may form salt bridges or polar interactions resulting in improved stability of the receptor. Finally, the double mutant $A98^{2.49}C/A140^{3.38}C$ can form a disulfide bridge between TM2 and TM3, apparently fixing the inactive conformation of the receptor. The latter observation is corroborated by the highest differential in thermostability between antagonist and agonist bound states measured for the combination construct containing the $A98^{2.49}C/A140^{3.38}C$ mutant (*Figure 6*).

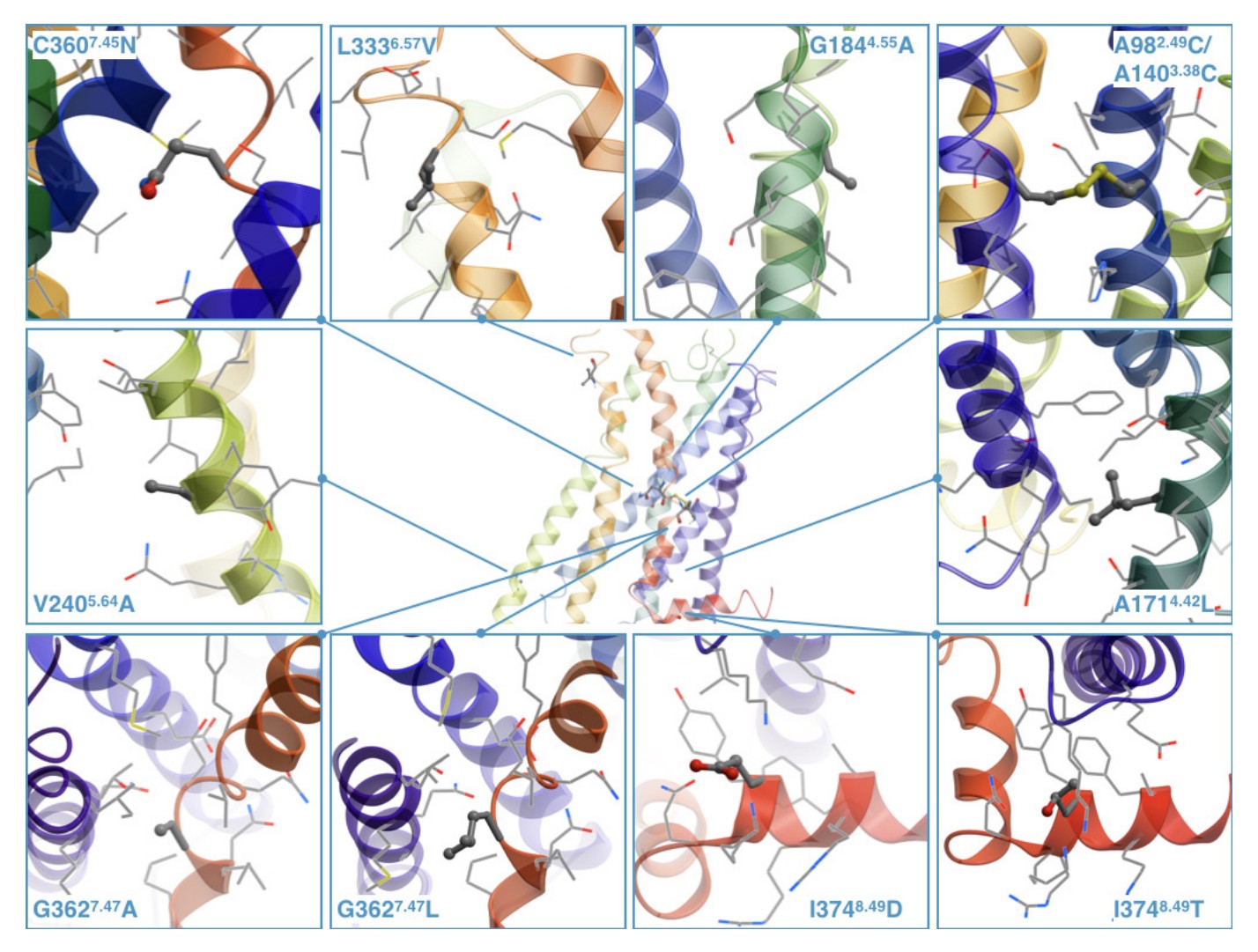

**Figure 7.** The stabilizing point mutations modeled in the structure of the ritanserin-bound 5-HT$_{2C}$ receptor (the ligand is not shown). Each mutated residue and its neighboring residues are represented as sticks and wires, respectively.
DOI: https://doi.org/10.7554/eLife.34729.012

## Discussion

Here we introduced a new comprehensive CompoMug approach for the prediction of stabilizing mutations in GPCRs and demonstrated its first application. In the 5-HT$_{2C}$ case, CompoMug achieved a very high success rate of 25% for mutations with substantial, more than 1.5°C, improvement in Tm values, which is 5 to 15 times higher than corresponding hit rates found in the experimental alanine mutagenesis approach for the adenosine or endothelin type B receptors (*Lebon et al., 2011*; *Okuta et al., 2016*). The CompoMug predictions resulted in the discovery of 10 new stabilizing mutations and enabled the structure determination of the 5-HT$_{2C}$ receptor in complexes with an agonist and an antagonist. Importantly, our first results suggest that each module of CompoMug is important and can additively contribute to the discovery of key stabilizing mutations. Below we discuss strengths and limitations of the individual CompoMug modules and how they can be further improved with an accumulation of structural and mutation data for the GPCR family.

*Knowledge-based* is the most established and straightforward approach, which directly copies some of the well-described mutations that already proved beneficial in crystallization of several GPCRs. However, while the mutations in the knowledge-based list are considered the most

transferable at least within class A GPCR, they still often fail to give any substantial gain in stability as it happened in the 5-HT$_{2C}$ case for the D99$^{2.50}$N and M143$^{3.41}$W mutations tested. Moreover, even in those cases when some of such known transferable mutants are beneficial, they may be not sufficient to get the optimal crystallization construct, nor to drive thermostability to the point required for structure-based drug design applications. Ongoing accumulation of mutation results for more GPCRs, e.g. discovered by other modules, may allow to add some additional mutations to the knowledge-based module. For example, X$^{3.41}$- > W mutation was initially discovered based on a sequence analysis and structure-based energy evaluations (*Roth et al., 2008*). At the same time, the recently discovered mutations in the functionally conserved sites (*Table 1*), e.g. in the P-I-F motif, DRY motif or sodium site (*Katritch et al., 2014*), apparently remove the key 'gears' from the activation mechanism, thus confining receptor to the inactive state and reducing conformational heterogeneity of the system, which can be beneficial for both thermostabilization and crystallization. In this respect, the sodium site residues provide the most opportunities, as it has several highly conserved sodium- and water-coordinating side chains that can be mutated.

*The sequence-based module* is fast and does not require structural knowledge. More than half of the tested point mutations (23 out of 39) came from the sequence-based module, with 6 of them yielding increased thermostability of the receptor, and 8 of them showing neutral effect. One of the advantages of this module is that mutation candidates have a lower probability to damage the receptor, because the candidate amino acid naturally occurs in other GPCRs. Moreover, in the particular case of 5-HT$_{2C}$ receptor, the highest improvement in thermostability was observed for the C360$^{7.45}$N point mutation predicted by the sequence-based module. Similar idea of using deviations in the residue conservation pattern as a potential target for mutation has been used by Chen et al to stabilize a variant of the ADRB1 receptor (*Chen et al., 2012*), though its application was limited to the cases of unusual polar or charged residues. In this study, we used an empirical score that is applicable to all types of residues and allows to quantify the predictions (see *Equation 1*). The score can be further optimized, for example, one can add specific weights with respect to MSAs, e.g. species variation may have a higher impact on the score compared to the common GPCR branch variation. A regression analysis required to adjust the optimal weights for the alignments, however would require further accumulation of additional stability data for GPCRs.

*The structure-based module* employs detailed information about residue interactions, and can potentially be highly predictive. In the 5-HT$_{2C}$ test case, nine tested point mutations were selected using the structure-based module, and two of them (I374$^{8.49}$D and A98$^{2.49}$C/A140$^{3.38}$C) increased the apparent melting temperature of the receptor. Importantly, the structure-based module can be very effective in differentiating between active and inactive state stabilizing conformations, and indeed the A98$^{2.49}$C/A140$^{3.38}$C disulfide bond 5-HT$_{2C}$ was shown to exclusively stabilize antagonist-bound complexes, but destabilize an agonist-bound complex (*Figure 6*). Previously, possibility to differentially stabilize GPCRs in agonist or antagonist-bund states was shown by experimental alanine mutagenesis study for the A$_{2A}$ receptor (*Magnani et al., 2008*). The structure-based module can do it very effectively, however, it requires a high-quality structural model, due to the high sensitivity of disulfide bonds, ionic locks and the corresponding energy terms to the receptor conformation. An increasing structural coverage of GPCR family, including both active- and inactive-state conformations will, in turn, allow more accurate models, improving performance and applicability of the structure-based module.

The *machine-learning-based* (ML) module allows to identify stabilizing point mutations overlooked by the other modules, since it uses complex feature vectors trained on previously obtained experimental data on mutations, rather than pure sequence or structural information. The machine learning, however, makes the resulting point mutations to be more difficult to interpret. We used eight point mutations selected by this module, of which four point mutations also showed high scores in the sequence-based module. Three of the ML point mutations improved the thermostability of the receptor. This module critically depends not only on the structural model, but also on the training set. For the current study, we used the available alanine mutations data, thus, its prediction power for the residues other than alanine can be limited. This situation will improve with an accumulation of novel stability data, including both experimental results coming from full receptor scanning and incorporation of mutations predicted by CompoMug in more than a dozen GPCRs.

By design, the four CompoMug modules are based on different principles and use different types of input information, so they are expected to complement each other, rather than overlap. Indeed,

our results suggest that most of the successful mutations for 5-HT$_{2C}$ were predicted by only one of the modules, and overlap in module predictions did not correlate with improved chances for the successful mutation, at least in this test case. Moreover, some of the trends in different modules can be opposite. For example, some of the specific knowledge-based mutations of highly conserved residues (e.g. D$^{2.50}$N), can render the receptor dysfunctional, but at the same time help its stability (*White et al., 2018*). On the other hand, the sequence-based module is based on a premise that residues deviating from the local conservation pattern are likely to be destabilizing. There is no contradiction here, as the evolutionary selection acted differently on different GPCR sites, in some positions selecting for general stability, but in other positions preserving conformationally unstable, but functionally critical residues.

The CompoMug tool is being applied to a number of GPCR targets, showing consistently high hit rates and helping structure determination of several GPCRs, including non-class A receptors (unpublished results). In principle, structure-based and sequence-based modules of CompoMug in their current form can be also applied to other membrane proteins families. The feature set of the machine-learning-based module may also have a more general utility, however, the model would require retraining on mutation stability data for the target family, where available.

## Conclusions

In this study, we present CompoMug - a computational tool to predict stabilizing point mutations in GPCRs. The four modules of CompoMug synergistically use different types of information on known transferable mutations, natural sequence variations, structural interactions, and machine learning of a large dataset of GPCR mutations, respectively, to maximize success rate of predictions. Applied to the 5-HT$_{2C}$ receptor, CompoMug helped us to identify as many as 10 stabilizing mutations (25% hit rate), supporting the importance of all four modules. One of the predicted mutations, C360$^{7.45}$N, improved the apparent melting temperature of the apo 5-HT$_{2C}$ receptor by 8.8 ± 1.3°C. Moreover, a triple mutant C360$^{7.45}$N, G362$^{7.47}$A, A171$^{4.42}$L had its thermostability improved by as much as ~13°C, as compared to the base construct apo receptor. Moreover, this C360$^{7.45}$N mutation in the optimal fusion construct yielded crystal structures of the 5-HT$_{2C}$ receptor in two distinct conformations, agonist-bound active like and antagonist-bound inactive. CompoMug is being applied to other receptors of the GPCR family, and performance of its modules can be further improved via the feedback loop with newly generated experimental data.

## Acknowledgements

This work was supported by the National Institute of Health grant P01DA035764 (VK, VC, RCS) Russian Science Foundation grant 16-14-10273 (VC), grant of the President of the Russian Federation MK-5279.2018.4 (PP), the Russian Ministry of Education and Science and Moscow Institute of Physics and Technology grant for visiting professors (VK), Ministry of Science and Technology of China grants 2014CB910400 and 2015CB910104, the National Natural Science Foundation of China grant 31330019 (ZJL), and GPCR Consortium research funds. We thank the Shanghai Municipal Government and ShanghaiTech University for financial support, as well as the Cloning, Cell Expression, and Protein Purification Core Facilities of iHuman Institute for their support.

## Additional information

### Competing interests

Petr Popov, Vsevolod Katritch: Filed a U.S. provisional patent application (serial no. 62/644,008) for CompoMug algorithms. The other authors declare that no competing interests exist.

### Funding

| Funder | Grant reference number | Author |
| --- | --- | --- |
| National Institutes of Health | P01DA035764 | Raymond C Stevens<br>Vadim Cherezov<br>Vsevolod Katritch |

| Russian Science Foundation | 16-14-10273 | Vadim Cherezov |
| --- | --- | --- |
| National Natural Science Foundation of China | 31330019 | Zhi-Jie Liu |
| Ministry of Science and Technology of China | 2014CB910400 | Zhi-Jie Liu |
| Ministry of Science and Technology of China | 2015CB910104 | Zhi-Jie Liu |

The funders had no role in study design, data collection and interpretation, or the decision to submit the work for publication.

### Author contributions

Petr Popov, Conceptualization, Resources, Data curation, Software, Formal analysis, Validation, Investigation, Visualization, Methodology, Writing—original draft, Writing—review and editing; Yao Peng, Data curation, Formal analysis, Validation, Investigation, Visualization, Methodology, Writing—review and editing; Ling Shen, Validation, Investigation, Visualization, Writing—review and editing; Raymond C Stevens, Conceptualization, Resources, Funding acquisition, Writing—review and editing; Vadim Cherezov, Conceptualization, Resources, Funding acquisition, Validation, Project administration, Writing—review and editing; Zhi-Jie Liu, Resources, Funding acquisition, Validation, Project administration, Writing—review and editing; Vsevolod Katritch, Conceptualization, Resources, Software, Supervision, Funding acquisition, Validation, Investigation, Methodology, Writing—original draft, Project administration, Writing—review and editing

### Author ORCIDs

Vsevolod Katritch (iD) http://orcid.org/0000-0003-3883-4505

### Decision letter and Author response

Decision letter https://doi.org/10.7554/eLife.34729.018
Author response https://doi.org/10.7554/eLife.34729.019

## Additional files

### Supplementary files

• Supplementary file 1. Top 40 CompoMug predictions for the three benchmark receptors. The true hits from the corresponding studies are highlighted green
DOI: https://doi.org/10.7554/eLife.34729.013

• Supplementary file 2. List of sequences used to construct five MSAs for $5HT_{2C}$ in the Sequence-based module.
DOI: https://doi.org/10.7554/eLife.34729.014

• Supplementary file 3. Key resources table.
DOI: https://doi.org/10.7554/eLife.34729.015

• Transparent reporting form
DOI: https://doi.org/10.7554/eLife.34729.016

### Data availability

All data generated in this study are included in the manuscript and supporting files. Source data for Figure 6 are provided in Supplementary Information table. CompoMug modules are available on GitLab (https://gitlab.com/pp_lab/CompoMug.git)

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
