## [Decision Letter]

Thank you for submitting your article "Rational Design of Thermostabilizing Point Mutations for G-Protein Coupled Receptors" for consideration by *eLife*. Your article has been reviewed by three peer reviewers, and the evaluation has been overseen by a Reviewing Editor and Arup Chakraborty as the Senior Editor. The following individual involved in review of your submission has agreed to reveal his identity: Dmitry Veprintsev (Reviewer #3).

The reviewers have discussed the reviews with one another and the Reviewing Editor has drafted this decision to help you prepare a revised submission.

Summary:

The manuscript describes a computational approach to the thermostabilisation of GPCRs. Prediction of thermo-stabilizing mutations is highly desirable because GPCRs that cannot be crystallized, are probably too unstable, and adding thermo-stabilizing mutations would make structure determination possible. The authors use the 5HT2c receptor as their target, because the structure had not been determined, and also there are close 5HT receptor homologs whose structures have been published (and therefore provide the best possible start for computational modelling). The program devised (CompoMug) is unique in that it takes into account four sources of information, whereas previous computational approaches normally considered only one. The four modules, knowledge-based, sequence-based, structure-based and machine learning, independently predicted point mutations that would thermo-stabilize the 5HT2c receptor. Sometimes, more than one module predicted the same mutation, but usually there was little overlap between the predictions. A total of 39 mutations were computationally predicted as being thermo-stabilizing, so these were expressed and purified, and the thermo-stability tested experimentally in the presence or absence of various ligands. Ten of the mutations (25%) were found to be thermo-stabilizing and were combined to make a triple mutant that was highly expressed and stable, and the structure was subsequently determined (accompanying manuscript).

There are two key features that sets this paper apart from previously published papers on predicting thermo-stabilizing mutations. Firstly, this manuscript considers and compares four different sources of information for predicting thermo-stabilizing mutations. Secondly, the predicted mutations were shown to be useful in generating a structure (reported in a separate paper). We appreciate that other GPCR structures have been published containing thermo-stabilizing mutations, but in many instances the authors have been less than candid about the approaches used to predict the mutations.

Essential revisions:

1) There are two numbers quoted for improved thermo-stability that are used throughout the paper to extol the virtues of the approaches developed, namely a 9˚C increase for a single mutation (C360N) and a 19˚C increase for a triple mutant. It appears that these numbers came from the graph in Figure 6 where the ΔTm is plotted for each of the constructs. However, the comparison is being made to the apo receptor, so the bar that gives the 9˚C increase is a C360N mutant receptor bound to ritanserin. The apo receptor containing the C360N mutation appears to have a ΔTm of 4˚C. However, this does not correspond to the data in Table 2 where the same mutation is shown with an apparent Tm increase of 8.6˚C. There cannot be two different numbers given for the same data. The 19˚C increase in apparent Tm seems to be of the mesulergine-bound mutant compared to the apo receptor. If it was compared to the wild type receptor bound to mesulergine, then the Tm difference would be about 5˚C lower. If the apo receptor is compared, the triple mutant would be about 10˚C more stable than the parental receptor. This graph needs to be re-drawn so that the increase in Tm is calculated between the parental receptor and mutant receptor when bound to the same ligand. This will give a true reflection on the improvement in apparent Tm. Secondly, all the actual Tms measured to create the data in Figure 6 must be included as a table in supplementary (or in the text). It is still worth putting in the bar graph as depicted in Figure 6, because it shows the absolute scale of thermo-stability and is therefore most useful in assessing the possibility of crystallizing a given mutant with a particular ligand. Error bars should also be put on both graphs.

2) The claims of Tm increases throughout the manuscript must be changed in the light of point 1.

3) The computational methods should be made available as scripts and/or web-server so that others can apply them to systems of interest.

4) It was unclear whether the method is transferable to other GPCRs. For generality, we recommend: (a) that the authors "retrospectively" predict stabilizing mutations used in crystal structures published by Heptares, and/or (b) that the authors implement their computational method (no additional experiments needed) to at least five other GPCRs of known structure and provide tables of suggested point mutations. Without this, the paper stands on only one example, although it claims a general method. We stress this point because two of the four modules the authors implement (knowledge based and structure based) may substantially narrow the scope.

5) In addition, the authors should include a brief discussion of potential applications of this algorithm to other classes of α-helical membrane proteins.

6) The authors claim that 25% hit rate is high. What are they comparing to? Alanine scan is obviously much lower, but structure based and especially consensus-based approaches have shown success rates of ~50% in soluble proteins.

7) Equation 1 is difficult to understand. It's essentially a way to quantify whether positions diverge from the family sequence consensus. We recommend carefully rephrasing the explanation of this equation taking care to correctly label the indices for each term (for instance, it seems that C_max_ should be Ckmax).

8) The sequence alignment can have a dramatic effect on prediction success, especially in divergent sequence families like GPCRs. Could the authors provide more details about sequence cutoffs and sources of information for the sequences? Ideally, they would provide the sequence ID of each sequence that went into each of the MSAs they used.

---

## [Author Response]

Essential revisions:

1) There are two numbers quoted for improved thermo-stability that are used throughout the paper to extol the virtues of the approaches developed, namely a 9˚C increase for a single mutation (C360N) and a 19˚C increase for a triple mutant. It appears that these numbers came from the graph in Figure 6 where the ΔTm is plotted for each of the constructs. However, the comparison is being made to the apo receptor, so the bar that gives the 9˚C increase is a C360N mutant receptor bound to ritanserin. The apo receptor containing the C360N mutation appears to have a ΔTm of 4˚C. However, this does not correspond to the data in Table 2 where the same mutation is shown with an apparent Tm increase of 8.6˚C. There cannot be two different numbers given for the same data. The 19˚C increase in apparent Tm seems to be of the mesulergine-bound mutant compared to the apo receptor. If it was compared to the wild type receptor bound to mesulergine, then the Tm difference would be about 5˚C lower. If the apo receptor is compared, the triple mutant would be about 10˚C more stable than the parental receptor. This graph needs to be re-drawn so that the increase in Tm is calculated between the parental receptor and mutant receptor when bound to the same ligand. This will give a true reflection on the improvement in apparent Tm. Secondly, all the actual Tms measured to create the data in Figure 6 must be included as a table in supplementary (or in the text). It is still worth putting in the bar graph as depicted in Figure 6, because it shows the absolute scale of thermo-stability and is therefore most useful in assessing the possibility of crystallizing a given mutant with a particular ligand. Error bars should also be put on both graphs.

Thank you for pointing to the apparent inconsistency in data presentation on Figure 6, where we initially used a wrong baseline value of Tm for the calculations. We carefully analyzed the data for the baseline construct (no mutations), and confirmed the baseline Tm = 50.4 ± 0.8°C, as listed in Table 2. The data also confirms accurate measurement of the most stabilizing mutant C360^7.45^N as 59.2 ± 0.5°C, resulting in the apparent ΔTm for this mutant 8.8 ± 1.3 °C. At the same time the baseline adjustment results in higher ΔTm =13°C for the triple mutant, and total gain for the triple mutant and antagonist stabilization as ΔTm =21°C. While initial apo receptor assessment in Table 2 was done in triplicate, the ligand assessment data in Figure 6 was performed only once due to limited amounts of the ligands. The standard error of these measurements can be estimated as ~1.2°C, as derived for the same assay in Table 2. Though the measurement of the apparent Tm for the most stabilizing mutation C360^7.45^N in a separate experiment shown in Figure 6 is somewhat an outlier (ΔTm=57.1 °C), it is still close to standard error of Tm measurement in Table 2.

Following reviewers’ suggestions, the updated Figure 6 now shows the Tm levels of the WT-construct in complexes with ligands in pale colors, the additional stabilizing effect of mutations in bright colored bars, to show both effects individually. Also, as reviewers suggested, we added Figure 6—source data 1 which reports all Tm values shown in Figure 6. We made corresponding changes in the Abstract, text of the manuscript, and the caption of Figure 6.

2) The claims of Tm increases throughout the manuscript must be changed in the light of point 1.

The Tm values in the text were adjusted according to changes described in answer #1.

3) The computational methods should be made available as scripts and/or web-server so that others can apply them to systems of interest.

We agree with the reviewers that the CompoMug tools should be made available for use by other researchers and we carefully considered ways to maximize its utility by GPCR community. We believe that the best option for making it conveniently accessible is to implement as a web service, and we have a collaboration in place with GPCRdb to make it accessible and compatible as a GPCRdb tool within the next 6-8 months. This, however, is a separate project beyond the scope of this article, and requires rewriting some part of the code for server implementation. In the current version, some of the scripts rely on commercially licensed software and are not ready for distribution.

4) It was unclear whether the method is transferable to other GPCRs. For generality, we recommend: (a) that the authors "retrospectively" predict stabilizing mutations used in crystal structures published by Heptares, and/or (b) that the authors implement their computational method (no additional experiments needed) to at least five other GPCRs of known structure and provide tables of suggested point mutations. Without this, the paper stands on only one example, although it claims a general method. We stress this point because two of the four modules the authors implement (knowledge based and structure based) may substantially narrow the scope.

a) As reviewers suggested, we performed an initial retrospective training set benchmarking with the results of previous alanine scanning experiments, and added it the beginning of the Results section. We emphasize, though, that this benchmarking is limited to alanine scan, which is only a small fraction of CompoMug predictions, and therefore employs only two out of four CompoMug modules. The main focus of the paper remains on the prospective test for 5HT_2C_.

b) We do have prospective CompoMug predictions for more than five other GPCRs. Moreover, crystal structures for at least 3 new GPCRs are currently being solved by our collaborators using mutations derived with CompoMug. Terms of our collaboration, however, do not allow to reveal mutations or even the identity of the targets until the structure is published, which is likely to take between 8 and 12 month. We believe it is important to publish the current methodology now, and to be able to refer to the CompoMug method in the upcoming publications of new GPCR structures. To comment on the successful transferability of CompoMug method to multiple GPCR targets, we added the following text to the end of Discussion section: “The CompoMug tool is being applied to a number of GPCR targets, showing consistently high hit rates and helping structure determination of several GPCRs, including non-class A receptors (unpublished results).”

5) In addition, the authors should include a brief discussion of potential applications of this algorithm to other classes of α-helical membrane proteins.

We added the following discussion paragraph at the end of Discussion section:

“In principle, structure-based and sequence-based modules of CompoMug in their current form can be also applied to other membrane proteins families. The feature set of the machine-learning-based module may also have a more general utility, however, the model would require retraining on mutation stability data for the target family, where available.”

6) The authors claim that 25% hit rate is high. What are they comparing to? Alanine scan is obviously much lower, but structure based and especially consensus-based approaches have shown success rates of ~50% in soluble proteins.

We compared with the success rate of the alanine scanning for GPCRs. For example, in a classical for the A2A receptor only 16 out of 315 (~5%) were reported to increase Tm by more than 1.5 °C (Table 1 in Lebon et al., 2011). The alanine scanning for the ETB receptor gave even lower hit rates: 5 out of 297 (1.5%) mutations were reported to increase Tm by 1.7°C or higher (Okuta et al., 2016). Thus, the CompoMug’s success rate of 25% is about 5 to 15 times higher, as compared to the alanine scanning on several known GPCRs examples. The first paragraph of the Discussion has been adjusted accordingly.

*7) Equation 1 is difficult to understand. It's essentially a way to quantify whether positions diverge from the family sequence consensus. We recommend carefully rephrasing the explanation of this equation taking care to correctly label the indices for each term (for instance, it seems that C_max_ should be*
Ckmax).

We corrected the Equation 1 and rephrased its explanation (subsection “Sequence-based module”).

8) The sequence alignment can have a dramatic effect on prediction success, especially in divergent sequence families like GPCRs. Could the authors provide more details about sequence cutoffs and sources of information for the sequences? Ideally, they would provide the sequence ID of each sequence that went into each of the MSAs they used.

We added details regarding the MSA construction in the text (subsection). In addition, we prepared Supplementary file 2 with the sequence IDs used to construct each of the five alignments.